# Molecular basis for N-terminal alpha-synuclein acetylation by human NatB

**Sunbin Deng[1,2], Buyan Pan[1], Leah Gottlieb[1,2], E James Petersson[1,3], Ronen Marmorstein[1,2,3]\***

[1]Department of Chemistry, University of Pennsylvania, Philadelphia, United States; [2]Abramson Family Cancer Research Institute, Perelman School of Medicine, University of Pennsylvania, Philadelphia, United States; [3]Department of Biochemistry and Biophysics, Perelman School of Medicine, University of Pennsylvania, Philadelphia, United States

**Abstract** NatB is one of three major N-terminal acetyltransferase (NAT) complexes (NatA-NatC), which co-translationally acetylate the N-termini of eukaryotic proteins. Its substrates account for about 21% of the human proteome, including well known proteins such as actin, tropomyosin, CDK2, and α-synuclein (αSyn). Human NatB (hNatB) mediated N-terminal acetylation of αSyn has been demonstrated to play key roles in the pathogenesis of Parkinson's disease and as a potential therapeutic target for hepatocellular carcinoma. Here we report the cryo-EM structure of hNatB bound to a CoA-αSyn conjugate, together with structure-guided analysis of mutational effects on catalysis. This analysis reveals functionally important differences with human NatA and *Candida albicans* NatB, resolves key hNatB protein determinants for αSyn N-terminal acetylation, and identifies important residues for substrate-specific recognition and acetylation by NatB enzymes. These studies have implications for developing small molecule NatB probes and for understanding the mode of substrate selection by NAT enzymes.

**\*For correspondence:** marmor@upenn.edu

**Competing interests:** The authors declare that no competing interests exist.

## Introduction

N-terminal acetylation (NTA) is an irreversible protein modification that predominantly occurs co-translationally on eukaryotic nascent chains emerging from the ribosome exit tunnel (*Aksnes et al., 2019*). This modification occurs on ~80% of human proteins (*Aksnes et al., 2016*) and impacts several protein functions including complex formation (*Scott et al., 2011*; *Monda et al., 2013*; *Yang et al., 2013*; *Arnaudo et al., 2013*; *Gao et al., 2016*), protein localization (*Behnia et al., 2004*; *Setty et al., 2004*; *Dikiy and Eliezer, 2014*; *Miotto et al., 2015*; *Forte et al., 2011*), and the N-end rule for protein degradation (*Hwang et al., 2010*; *Shemorry et al., 2013*; *Park et al., 2015*). NTA is catalyzed by a group of enzymes called N-terminal acetyltransferases (NATs), which belong to the Gcn5-related N-acetyltransferase (GNAT) family of enzymes. In humans, seven NATs have been identified to date: NatA, NatB, NatC, NatD, NatE, NatF, and NatH (*Aksnes et al., 2019*). NatA, NatB and NatC are responsible for the majority of NTA (*Starheim et al., 2012*). NatA and NatB are heterodimeric complexes, each with a distinctive catalytic subunit (NAA10 in NatA and NAA20 in NatB; *Starheim et al., 2008*) and a unique auxiliary subunit (NAA15 in NatA and NAA25 in NatB; *Arnesen et al., 2005*). NatC is a heterotrimeric complex that contains a catalytic subunit (NAA30) and two auxiliary subunits (NAA35 and NAA38; *Starheim et al., 2009*). Structural and biochemical studies of NatA reveal that the auxiliary subunit plays an important allosteric role in substrate binding specificity and catalysis (*Liszczak et al., 2013*) and also contributes to ribosome binding during translation (*Magin et al., 2017*; *Knorr et al., 2019*). NatA, NatB, and NatC exhibit distinguishable substrate specificity mainly toward the first two residues of the nascent chain N-termini. NatA accounts for 38% of the human N-terminal acetylome, acetylating small N-terminal

residues after removal of the initial methionine residue (*Arnesen et al., 2009*), while NatB modifies N-termini with sequences of MD-, ME-, MN-, and MQ- (*Van Damme et al., 2012*; *Van Damme et al., 2012*). NatC/E/F have overlapping substrates, acting on N-terminal methionine when it is followed by several residues excluding D, E, N, and Q (*Arnesen et al., 2009*; *Van Damme et al., 2016*; *Tercero et al., 1993*; *Evjenth et al., 2009*; *Van Damme et al., 2011*; *Støve et al., 2016*; *Liszczak et al., 2011*). When another catalytic subunit (NAA50) binds to NatA, a dual enzyme complex NatE is formed (*Gautschi et al., 2003*; *Deng et al., 2020*), with catalytic crosstalk between NAA10 and NAA50 (*Deng et al., 2020*; *Deng et al., 2019*). We recently demonstrated that both NAA50, and a protein with intrinsic NatA inhibitory activity - Huntingtin-interacting protein K (HYPK; *Gottlieb and Marmorstein, 2018*; *Weyer et al., 2017*; *Arnesen et al., 2010*), can bind to NatA simultaneously to form a larger tetrameric complex (*Deng et al., 2020*). NatD and NatH are highly selective enzymes with restricted substrates, displaying activity toward H4/H2A and N-terminally processed actin, respectively (*Goris et al., 2018*; *Drazic et al., 2018*; *Song et al., 2003*). The unique localization of NatF to the Golgi membrane demonstrates NTA can occur post-translationally in some cases (*Aksnes et al., 2015*).

NatB is conserved from yeast to human in both complex composition and in its substrate specificity profile (*Starheim et al., 2012*). In *Saccharomyces cerevisiae*, the deletion of NatB subunits produces more severe phenotypes compared to the knockout of NatA or NatC subunits. Deletion of either NAA20 or NAA25 leads to similar phenotypes including slower growth rate, diminished mating, defects in actin cable formation, and aberrant mitochondrial and vacuolar inheritance (*Polevoda et al., 2003*). These observations suggest that the proper function of actin and tropomyosin requires NTA by the intact NatB complex (*Polevoda et al., 2003*). In humans, disruption of NatB (hNatB) by knockout leads to defects in proper actin cytoskeleton structure, cell cycle progression, and cell proliferation (*Starheim et al., 2008*; *Ametzazurra et al., 2008*; *Ametzazurra et al., 2009*; *Neri et al., 2017*). In addition, hNatB is upregulated in human hepatocellular carcinoma (*Ametzazurra et al., 2008*), where it has been suggested as a potential therapeutic target as silencing of this complex can block cell proliferation and tumor formation (*Neri et al., 2017*). hNatB-mediated NTA of α-synuclein (αSyn) has been shown to increase αSyn stability and lipid binding, and to reduce aggregation capacity (*Dikiy and Eliezer, 2014*; *Watson and Lee, 2019*; *Mason et al., 2016*; *Maltsev et al., 2012*; *Trexler and Rhoades, 2012*; *Fernández and Lucas, 2018*; *Fauvet et al., 2012*; *Kang et al., 2012*; *Iyer et al., 2016*). Since αSyn is a key protein in Parkinson's disease (PD; *Halliday et al., 2011*; *Spillantini et al., 1998*), hNatB might play an indirect role in PD pathogenesis in vivo as supported by a recent study (*Vinueza-Gavilanes et al., 2020*). It was also recently demonstrated that NTA of αSyn increases its propensity for lipid membrane binding without altering its structural properties of the bound state (*Runfola et al., 2020*).

Compared to the comprehensive structural and biochemical characterization of NatA (*Liszczak et al., 2013*; *Magin et al., 2017*; *Knorr et al., 2019*; *Deng et al., 2019*; *Gottlieb and Marmorstein, 2018*; *Weyer et al., 2017*), the study of NatB has been limited, particularly in humans. Recently, the crystal structure of *Candida albicans* (*Ca*) NatB bound to a bisubstrate CoA-peptide conjugate was determined, providing important insights into substrate specificity and NTA by caNatB (*Hong et al., 2017*). However, hNAA20 and hNAA25 share only ~40% and ~20% sequence identity with the *Candida albicans* homolog (*Hong et al., 2017*), respectively. Moreover, nearly all biological studies of NatB have been conducted in *Saccharomyces cerevisiae* (*Lee et al., 2014*; *Caesar et al., 2006*; *Singer and Shaw, 2003*), Arabidopsis (*Ferrández-Ayela et al., 2013*; *Huber et al., 2020*), mouse (*Ohyama et al., 2012*), and human (*Starheim et al., 2008*; *Ametzazurra et al., 2008*; *Ametzazurra et al., 2009*; *Neri et al., 2017*) as model organisms. As a result, the mode of human NatB-mediated catalysis and αSyn-specific NatB recognition remains unresolved. In this study, we report the 3.5 Å resolution cryo-electron microscopy (cryo-EM) structure of the ~130 KDa hNatB bound to a bisubstrate CoA-αSyn conjugate together with a structure-guided analysis of mutational effects on catalytic activity. This analysis reveals functionally important structural differences between hNaB and related NAT enzymes, as well as insights into the molecular mechanisms that define αSyn and related substrates that are recognized for hNatB-mediated N-terminal acetylation.

## Results

### hNatB is potently inhibited by a CoA-αSyn conjugate

While attempts to express recombinant hNatB in *E. coli* were unsuccessful, we found that overexpression of hNatB complex with full-length hNAA25 (residues 1–972) and C-terminally truncated hNAA20 (residue 1–163 out of 178 total residues) in baculovirus-infected Sf9 insect cells produced soluble protein that could be purified to homogeneity (*Figure 1A*). To evaluate the activity of the recombinant hNatB, we tested it against different peptide substrates. αSyn with an N-terminal sequence of 'MDVF' has been widely considered as an in vivo hNatB substrate (*Van Damme et al., 2012*; *Anderson et al., 2006*; *Ohrfelt et al., 2011*; *Theillet et al., 2016*). We, therefore, incorporated this sequence into a peptide substrate named 'MDVF' for an in vitro acetyltransferase assay ('MDVF' peptide sequence: $NH_2$-MDVFMKGRWGRPVGRRRRP-COOH). In agreement with in vivo studies (*Van Damme et al., 2012*; *Anderson et al., 2006*; *Ohrfelt et al., 2011*; *Theillet et al., 2016*), we observed that the purified recombinant hNatB was active against this 'MDVF' peptide, while no activity could be observed in the absence of either the enzyme or peptide (*Figure 1B*). hNatB also showed no observable activity if either the first residue 'M' or the first two residues 'MD' in this αSyn peptide substrate was removed ('DVFM' peptide sequence: $NH_2$-DVFMKGLRWGRPVG RRRRP-COOH; 'VFMK' peptide sequence: $NH_2$-VFMKGLSRWGRPVGRRRRP-COOH; *Figure 1B*), suggesting that peptide substrate recognition by NatB is highly dependent on the first two N-terminal residues. To further confirm the substrate specificity of hNatB, we tested it against several previously identified peptide substrates for other NATs ('SASE' peptide sequence (NatA-type): $NH_2$-SA SEAGVRWGRPVGRRRRP-COOH; 'MLRF' peptide sequence (NatC-type): $NH_2$-MLRFVTKRWGRPVG RRRRP-COOH; 'SGRG'/H4 peptide sequence (NatD-type): $NH_2$-SGRGKGGKGLGKGGAKRHR-

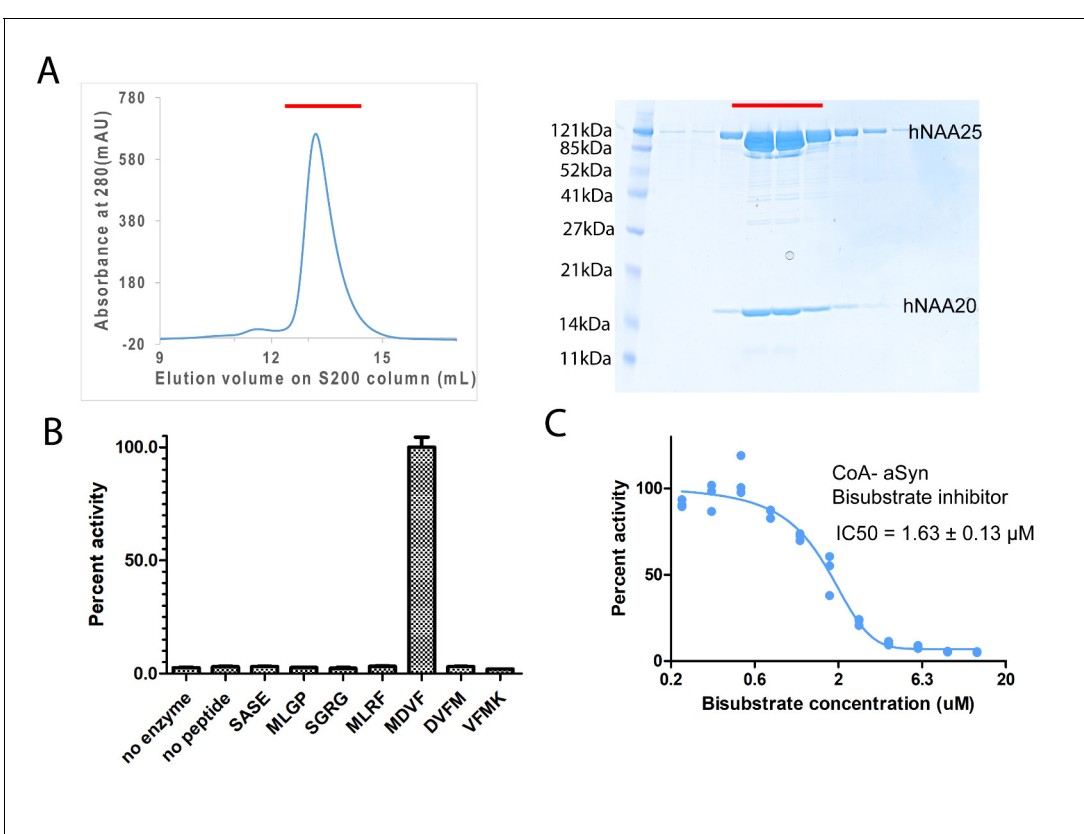

**Figure 1.** hNatB is active toward an α-Synuclein peptide and can be inhibited by a CoA-αSyn conjugate. (**A**) Gel filtration elution profile of hNatB, using a Superdex S200 column. Coomassie-stained SDS-PAGE of peak fractions is reproduced to the right of the chromatograms. (**B**) Comparison of hNatB activity toward different peptide substrates. All the activities are normalized to the activity of hNatB toward αSyn peptide (MDVF). (**C**) The dose-response curve corresponding to the titration of CoA-αSyn conjugate (CoA-MDVFMKGLSK) into hNatB acetyltransferase reactions. The calculated $IC_{50}$ value is indicated. Reactions were performed in triplicate; replicates are shown in the graph as vertical dots.

COOH; 'MLGP' peptide sequence (NatE-type): NH$_2$-MLGPEGGRWGRPVGRRRRP-COOH; *Figure 1B*). Consistent with previous results (*Van Damme et al., 2012*), hNatB is only active toward its unique canonical substrate type, displaying no overlapping activity toward other NAT substrates.

In order to understand the mechanism of hNatB substrate recognition, we synthesized a bi-substrate inhibitor in which the first 10 residues of αSyn are covalently linked to CoA (*Liszczak et al., 2013*) for enzymatic and structural studies. Half-maximum inhibitory concentration (IC$_{50}$) determinations revealed that this CoA-αSyn conjugate had an IC$_{50}$ of about 1.63 ± 0.13 μM (*Figure 1C*), significantly lower than the K$_m$ values we had determined for hNatB toward a 'MDVF' peptide (45.08 ± 3.15 μM) and acetyl-CoA (47.28 ± 5.70 μM) (*Table 1*).

## hNatB reveals potentially biologically significant structural differences with hNatA and caNatB

We performed single particle cryo-EM of hNatB in the presence of the CoA-αSyn conjugate. A 3.46 Å-resolution cryo-EM three-dimensional (3D) map was determined from 982,420 particles, selected from 5281 raw electron micrographs (*Table 2*, *Figure 2—figure supplement 1* and *Figure 2—figure supplement 2*). The central core region of the EM map contains excellent sidechain density with a local resolution of ~2.5 Å, particularly around the catalytic subunit, hNAA20.

Consistent with previous NAT structural studies, the atomic model of hNatB features a catalytic subunit, hNAA20, that adopts a canonical Gcn5-related N-acetyltransferase (GNAT) fold (*Deng and Marmorstein, 2020*; *Neuwald and Landsman, 1997*). Additionally, the model reveals that the auxiliary subunit hNAA25 is composed of a total of 39 α-helices among where the predicted first and second α-helices were built as poly-alanine due to a lack of resolvable sidechain density (*Figure 2—figure supplement 3*). The 39 α-helices can be roughly divided into three groups: an N-terminal region: α1-α8; a core region: α9-α29; and a C-terminal region: α30-α39 (*Figure 2A* and *Video 1*). The N-terminal region (residue 1–164) displays relatively weak EM density compared to other regions, suggesting that it is relatively flexible (*Figure 2—figure supplement 2*). The eight helices of the N-terminal region form four helical bundle tetratricopeptide repeat (TPR) motifs, which often participate in protein-protein interactions. While there are no visible contacts between the N-terminal TPR motifs and hNAA20, it is possible that this region participates in ribosome association, similar to the N-terminal region of the NAA15 auxiliary subunits of *Schizosaccharomyces pombe* (*Magin et al., 2017*) and *Saccharomyces cerevisiae* (*Knorr et al., 2019*) NatA. The 21 helices of the core region also form a number of TPR motifs, which come together to form a ring that completely wraps around and extensively contacts hNAA20 within its hollow center (*Figure 2A*). Indeed, the interaction between hNAA20 and the TPR motifs of this core region buries a total interface area of about 2300 Å$^2$. In the core region, it is noteworthy that there is a long α-helix (α28, ranging 30

**Table 1.** Catalytic parameter of wild-type hNatB and mutants.

| Substrate | Protein | K$_{cat}$ (min$^{-1}$) | K$_{cat}$ (normalized to WT) | K$_m$ (μM) | K$_m$ (normalized to WT) | K$_{cat}$/K$_m$ (normalized to WT) |
|---|---|---|---|---|---|---|
| Acetyl-CoA | WT | 9.25 ± 0.29 | 1 | 47.28 ± 5.70 | 1 | 1 |
| MDVF peptide | WT | 7.63 ± 0.14 | 1.0 | 45.08 ± 3.15 | 1.0 | 1.0 |
| | E25A | 8.31 ± 0.35 | 1.1 | 39.30 ± 6.62 | 0.87 | 1.3 |
| | Y27A | 16.73 ± 2.11 | 2.2 | 75.03 ± 33.26 | 1.7 | 1.3 |
| | H73A | 0.89 ± 0.14 | 0.12 | 54.76 ± 32.81 | 1.2 | 0.10 |
| | R84A | 13.86 ± 1.75 | 1.8 | 320.8 ± 96.7 | 7.1 | 0.25 |
| | R85A | 18.65 ± 1.02 | 2.4 | 109.6 ± 19.33 | 2.4 | 1.0 |
| | G87A | 14.86 ± 0.56 | 1.9 | 78.16 ± 10.25 | 1.7 | 1.1 |
| | N116A | 0.90 ± 0.06 | 0.12 | 39.86 ± 11.84 | 0.88 | 0.14 |
| | Y123A | 0.34 ± 0.03 | 0.045 | 9.04 ± 3.81 | 0.20 | 0.23 |
| | Y123F | 0.94 ± 0.06 | 0.12 | 42.43 ± 11.52 | 0.94 | 0.13 |
| | Y137A | 6.43 ± 0.23 | 0.84 | 53.67 ± 9.63 | 1.2 | 0.70 |
| | Y138A | 3.09 ± 0.25 | 0.40 | 44.23 ± 13.87 | 0.98 | 0.41 |

**Table 2.** Cryo-EM data collection, refinement, and validation statistics.

| | hNatB/CoA-αSyn complex EMD-21307 PDB: 6VP9 EMPIAR-10477 |
|---|---|
| Data collection and processing | |
| Magnification | 105,000 |
| Voltage (keV) | 300 |
| Electron exposure (e/Å$^2$) | 40 |
| Defocus range (μm) | −1.5 to −2.5 |
| Pixel size (Å) | 0.83 |
| Symmetry imposed | C1 |
| Initial particles (no.) | 1,927,675 |
| Final particles (no.) | 982,420 |
| Map resolution (Å) | 3.46 |
| FSC threshold | 0.143 |
| Map resolution range (Å) | 2.5–4.5 |
| Refinement | |
| Initial model used (PDB code) | - |
| Model resolution (Å) | 3.5 |
| FSC threshold | 0.5 |
| Model resolution range (Å) | - |
| Map sharpening $B$ factor (Å$^2$) | −191.177 |
| Model composition | |
| Non-hydrogen atoms | 8611 |
| Protein residues | 1077 |
| Ligands | 2 |
| $B$ factors (Å$^2$) | |
| Protein | 2.51/84.68/33.39 |
| Ligand | 17.07/19.91/19.74 |
| R.M.S. deviations | |
| Bonds lengths (Å) | 0.011 |
| Bond angles (°) | 1.105 |
| Validation | |
| MolProbity score | 1.73 |
| Clash score | 3.98 |
| Poor rotamers (%) | 0.55 |
| Ramachandran plot | |
| Favored (%) | 90.10 |
| Allowed (%) | 9.71 |
| Disallowed (%) | 0.19 |

residues) that traverses almost from one side of the complex to the other. The α28 helix closes the core ring structure, locking hNAA20 in position, and bridging the N- and C-terminal and regions. This is similar to the role played by α29-α30 of the hNAA15 auxiliary subunit of hNatA (*Figure 2A and B*). The C-terminal region features helices that bundle together to protrude out of the plane of the core ring structure at an angle of ~45° (*Figure 2B*).

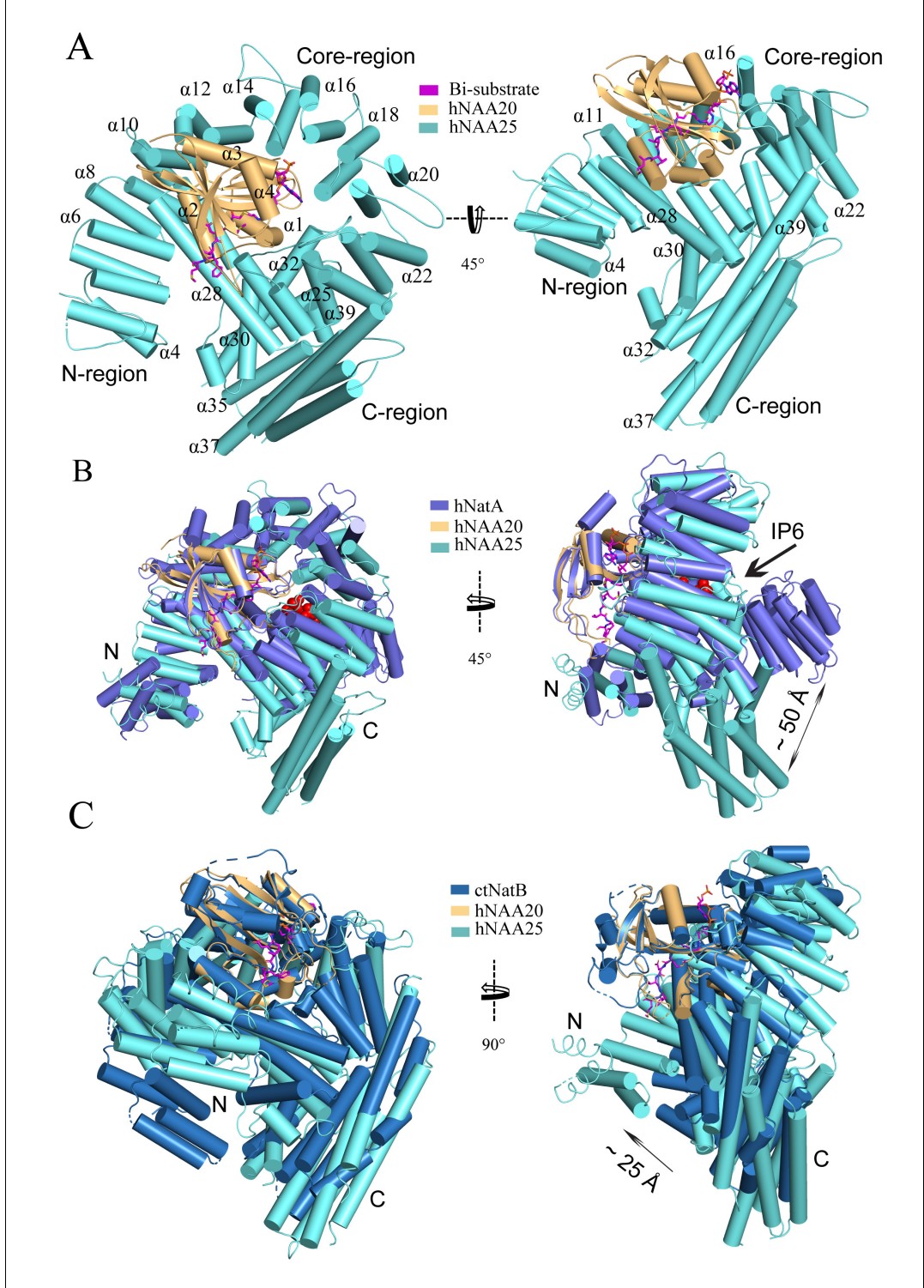

**Figure 2.** hNatB shows structural differences with hNatA and *C.albicans* NatB. (**A**) hNaa20 (light orange) and hNaa25 (cyan) are shown in cartoon. The CoA-αSyn conjugate inhibitor is shown in sticks and colored as magenta. The N- and C- terminal regions are indicated as 'N-region' and 'C-region', respectively. Some helices are as labeled. (**B**) hNaa20 (light orange) and hNaa25 (cyan) are shown overlapped with hNatA (marine blue, PDB: 6C9M). Small molecule IP6 bound to hNatA is shown as surface representation (red). (**C**) hNaa20 (light orange) and hNaa25 (cyan) are shown superimposed on *Ca*NatB (slate blue, PDB:5K04).

The online version of this article includes the following figure supplement(s) for figure 2:

*Figure 2 continued on next page*

*Figure 2 continued*

**Figure supplement 1.** Cryo-EM workflow and resolution of hNatB map.
**Figure supplement 2.** Representative micrograph of hNatB cryo-images and hNatB model fit-in map.
**Figure supplement 3.** Sequence alignment of NAA25 homologs.
**Figure supplement 4.** Structural comparison of hNatB and hNatA-HYPK.

The sequence identity of the catalytic and auxiliary subunits of hNatA and hNatB are 20% and 15%, respectively. To understand how this translates to key structural differences, we superimposed the crystal structure of hNatA (PDB: 6C9M) with our model. Between the catalytic subunits, there is a high degree of superposition (1.151 Å root-mean-square deviation [RMSD] over 105 common $C_\alpha$ atoms), except for an additional helix, α5, on the C-terminus of hNatA-NAA10, which is absent in hNatB-NAA20 (*Figure 2B*). Between the auxiliary subunits, the core and N-terminal regions of both hNatA-NAA15 and hNatB-NAA25 display similar topology, although a higher degree of deviation than the catalytic subunits. The core regions both wrap around their respective catalytic subunits (8.369 Å RMSD over 262 common $C_\alpha$ atoms), while the N-terminal regions jut off to the side (7.360 Å RMSD over 55 common $C_\alpha$ atoms). By contrast, the C-terminal regions of hNatA-NAA15 and hNatB-NAA25 diverge significantly from one another. For hNA25, the C-terminal region of hNAA25 is oriented toward its N-terminal region, while the C-terminal region of hNAA15 is positioned ~50 Å away from the relative position of the superimposed C-terminal domain of hNAA25 (*Figure 2B*). The positioning of hNAA25 may serve to promote hNAA25 intra-termini communication, which is similar to the interaction of hNatA and its regulatory protein HYPK (*Gottlieb and Marmorstein, 2018*). HYPK, which does not interact with hNatB, interacts with both the N- and C-terminal domains of hNatA-NAA15, potentially serving as bridge to enable closer communication between these two domains (*Figure 2—figure supplement 4*). Recent reports have described the role of the small molecule $IP_6$ (inositol hexakisphosphate) in hNatA activity, where it is found to act as 'glue' between the C-terminal and core domains in hNAA15 and hNAA10 via a series of hydrogen bonds and electrostatic interactions (*Gottlieb and Marmorstein, 2018*; *Cheng et al., 2019*). While no corresponding small molecules have been identified to play a similar role in hNatB, our model shows that this interaction is replaced by an extended loop that connects the α31 helix with the α32 helix of hNatB-NAA25. This loop, which is not present in hNatA-NAA15, appears to mediate hydrophobic interactions between hNatB-NAA25 and -NAA20, likely to serve a similar role as $IP_6$ (*Figure 2—figure supplement 4B*).

We also compared the structures from human and the previously described *C. albicans* NatB (*Ca*NatB, PDB: 5K18). Although the two superimposed structures revealed a high degree of structural conservation (NAA20: 0.698 Å RMSD over 125 common $C_\alpha$ atoms; NAA25 Core region: 3.267 Å RMSD over 266 common $C_\alpha$ atoms), the N-terminal region of hNatB-NAA25 appears to overlay more closely to hNatA-NAA15 than to *Ca*NatB-NAA25 (*Figure 2B and C*). Compared to *Ca*NAA25, the N-terminal regions of hNatB-NAA25 and hNatA-NAA15 are positioned more closely to the peptide substrate binding sites of the respective catalytic subunits. Based on the role that the N-terminal yeast NatA-Naa15p regions play in ribosome docking (*Magin et al., 2017*; *Knorr et al., 2019*), we propose that the relative shift in the position of the N-terminal regions of the human NAT auxiliary subunits, hNAA15 and hNAA25, may reflect a difference in the mechanism for ribosome association and co-translational NTA in *C. albicans* compared with humans. In addition, the overlay of C-terminal regions of hNAA25 and *Ca*NAA25 displays an RMSD of 15.960 Å over 133 common $C_\alpha$ atoms. We observe that the main difference that

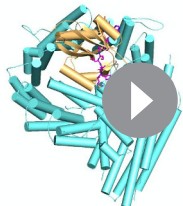

**Video 1.** Overall view of the NatB complex. hNaa20 (light orange) and hNaa25 (cyan) are shown in cartoon. The CoA-αSyn conjugate inhibitor is shown in sticks and colored as magenta.
https://elifesciences.org/articles/57491#video1

contributes to this deviation in this region is the length of helices.

## hNAA25 and hNAA20 make intimate interactions within hNatB

The hNatB/CoA-αSyn structure reveals an extensive interaction interface between the core region of the auxiliary hNAA25 and catalytic hNAA20 subunits (*Figure 3A*). The most intimate contact between the two proteins is made by the α28-α29 segment of hNAA25 and almost the entire length of hNAA20 α2, creating a large hydrophobic interface (*Figure 3B*). Residues that contribute to the interaction include Thr26, Gly28, Ile29, Pro30, Leu33, Gln34, Leu36, Ala37, His38, and Glu41 of

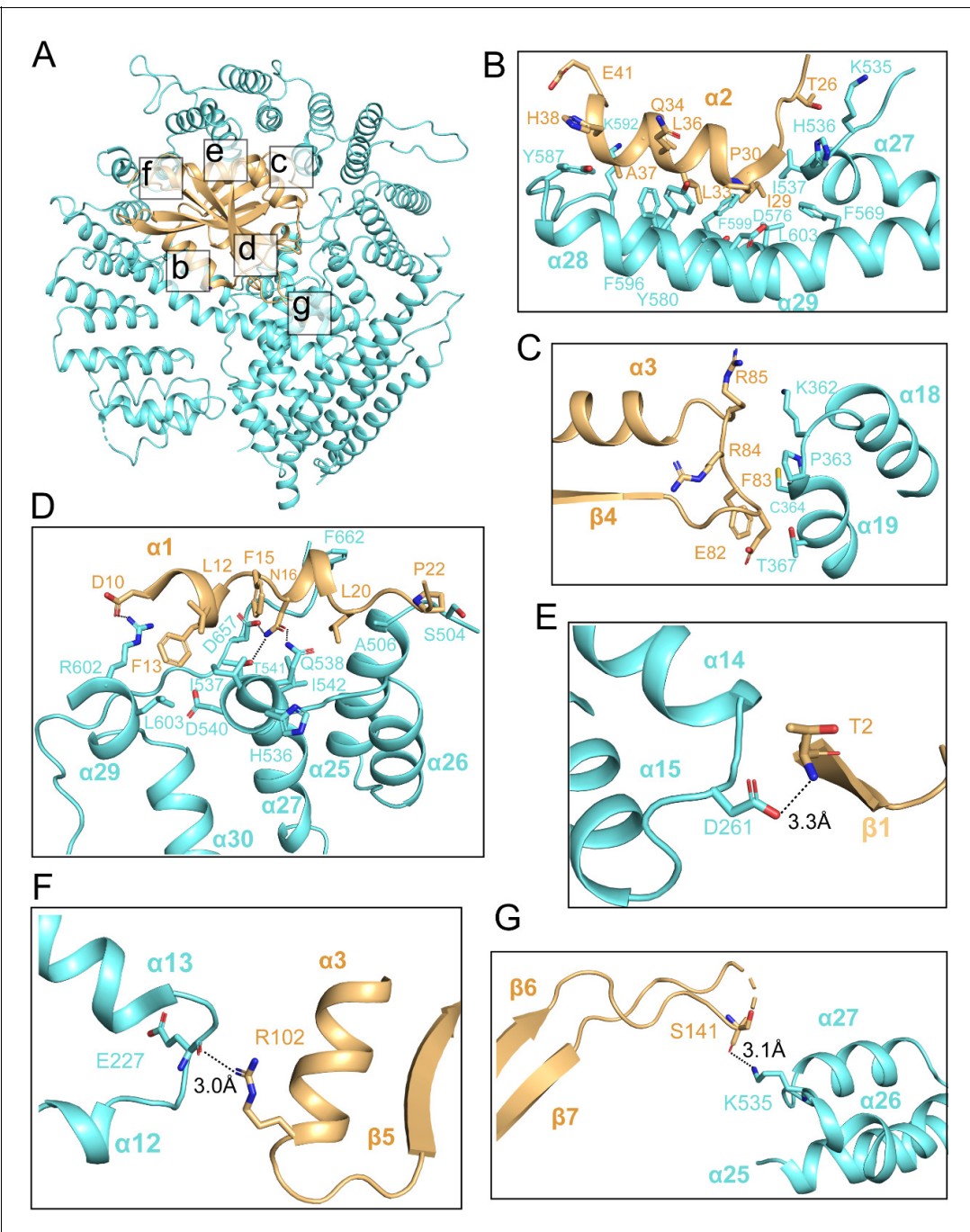

**Figure 3.** hNAA20 and hNAA25 make intimate interactions within hNatB. (A) hNAA20 (light orange) and hNAA25 (cyan) are shown in a cartoon with major associated interface denoted. (B–G) Zoom-in views of the hydrophobic interface regions as indicated in (A).

hNAA20 and Lys535, His536, Ile537, Phe569, Asp576, Thr577, Tyr580, Ala584, Tyr 587, Lys592, Phe596, Phe599, and Leu603 of hNAA25 (*Figure 3B*). Another region of interaction, involving predominantly van der Waals interaction occurs between the hNAA20 β4-α3 loop and the hNAA25 α18-α19 loop. Here residues Lys362, Pro363, Lys362,Thr367, Cys364, and Pro363 from hNAA25 and residues Arg85, Phe83, Glu82, and Arg84 from hNAA20 contribute to this small batch of hydrophobic interface (*Figure 3C*).

Additional intimate contacts between hNAA20 and hNAA25 are mediated between hNAA20 α1 helix and α26-α27 loop, and the hNAA25 α24-α25 loop, consisting of a mix of polar and non-polar interactions. A few hydrogen bonds are formed, centered around hNAA20-Asn16, between the sidechain nitrogen atoms of hNAA20-Asn16 and sidechain oxygen of hNAA25-Thr541, between the sidechain oxygen atom of hNAA20-Asn16 and sidechain nitrogen atom of hNAA25-Gln538, and between the sidechain nitrogen of hNAA20-Asn16 and the sidechain of hNAA25-Asp657 (*Figure 3D*). A salt bridge is found between the sidechain of hNAA20-Asp10 and the sidechain of hNAA25-Arg602. In addition, there are hydrophobic interactions between residues Phe15, Leu12, Phe13, Leu19, and Pro22 from hNAA20 and residues Ile537, Asp540, His536, Ala506, Phe662, Ser504, Ile542, and Leu603 from hNAA25 (*Figure 3D*).

Different sides of hNAA20 feature several potentially hNAA25-stabilizing polar interactions. Hydrogen bonds are formed between the Asp261 sidechain of the hNAA25 α14-α15 loop and the hNAA20-Thr2 backbone nitrogen atom (*Figure 3E*), between the Glu227 backbone carbonyl group of hNAA25 α12-α13 loop and the hNAA20-Arg102 sidechain (*Figure 3F*), and between the Lys535 sidechain from NAA25 α26-α27 loop and the hNAA20-Ser141 backbone carbonyl group (*Figure 3G*).

## hNatB makes specific interactions with the first 4 N-terminal residues of αSyn

In the cryo-EM map, density for the CoA-αSyn conjugate bisubstrate inhibitor is well resolved, allowing us to confidently model the CoA portion and the first 5 N-terminal residues (of 10 residues present) of the αSyn portion (*Figure 4A and B*, *Video 2*). Similar to other NATs, CoA enters the catalytic active site through a groove formed by α3 and α4 of the catalytic subunit, while the peptide substrate enters the active site on the opposite side of the catalytic subunit flanked by the α1–2 and β6-β7 loops (*Deng and Marmorstein, 2020*; *Figure 4A*). hNAA20 contains a conserved acetyl-CoA binding motif among NATs: $R_{84}R_{85}XG_{87}XA_{89}$ (*Figure 4—figure supplement 1*). Here we observe that the positively charged hNAA20-Arg85 interacts with the negatively charged 3'-phosphorylated ADP portion of CoA to form a salt bridge while Arg84 makes Van der Waals interactions (*Figure 4C and D*). A hydrogen-bonding network is formed mainly between the 5'-diphosphate group and backbone atoms of a few residues including Val79, Gly87, Ala89, and Ala90 (*Figure 4C and D*), and mediated by the sidechains of Arg85 and Gln125. The CoA molecule anchors to the binding pocket through a series of van der Waals contacts formed by residues Ser67, Val79, Leu77, Leu88, Val118, Met 122, and Tyr123 (*Figure 4D*).

Four N-terminal residues of αSyn participate in hNAA20 interactions. Anchoring of the αSyn peptide is mediated by protein hydrogen bonds with the backbone atoms of Met1 and Asp2 of αSyn. Hydrogen bonds are formed between the backbone N-H group of αSyn-Met1 and the backbone carbonyl group of hNAA20-Phe111, as well as the backbone carbonyl group of αSyn-Met1 with the sidechain of hNAA20-Tyr138. The backbone N-H and carbonyl of αSyn-Asp2 also form hydrogen bonds to the sidechain of Tyr27 and between the backbone carbonyl group of Asp2 and sidechain of hNAA20-Tyr27 (*Figure 4D*). Remarkably, hNAA20 contacts each of the first 4 N-terminal residue sidechains of αSyn via van der Waals interactions. The only sidechain that forms a hydrogen bond with hNAA20 is αSyn-Asp2, which hydrogen bonds with a hNAA20-His73 ring nitrogen and the hNAA20-Thr75 sidechain (*Figure 4D*). The more extensive van der Waals interactions include the following: αSyn-Met1 interacts with hNAA20 residues Glu25, Phe27, Tyr56, and Ala76; αSyn-Asp2 interacts with hNAA20 residues Tyr27, Thr75, His73, Phe111, and Tyr138; αSyn-Val3 interacts with hNAA20 residues Tyr137 and Tyr138; and αSyn-Phe4 interacts with hNAA20 residues Glu25 and Als140. αSyn-Met5 does not appear to make specific interactions (*Figure 4D*). Consistent with the importance of the residues that mediate αSyn binding, most of the residues are highly conserved from yeast to humans (*Figure 4—figure supplement 1*).

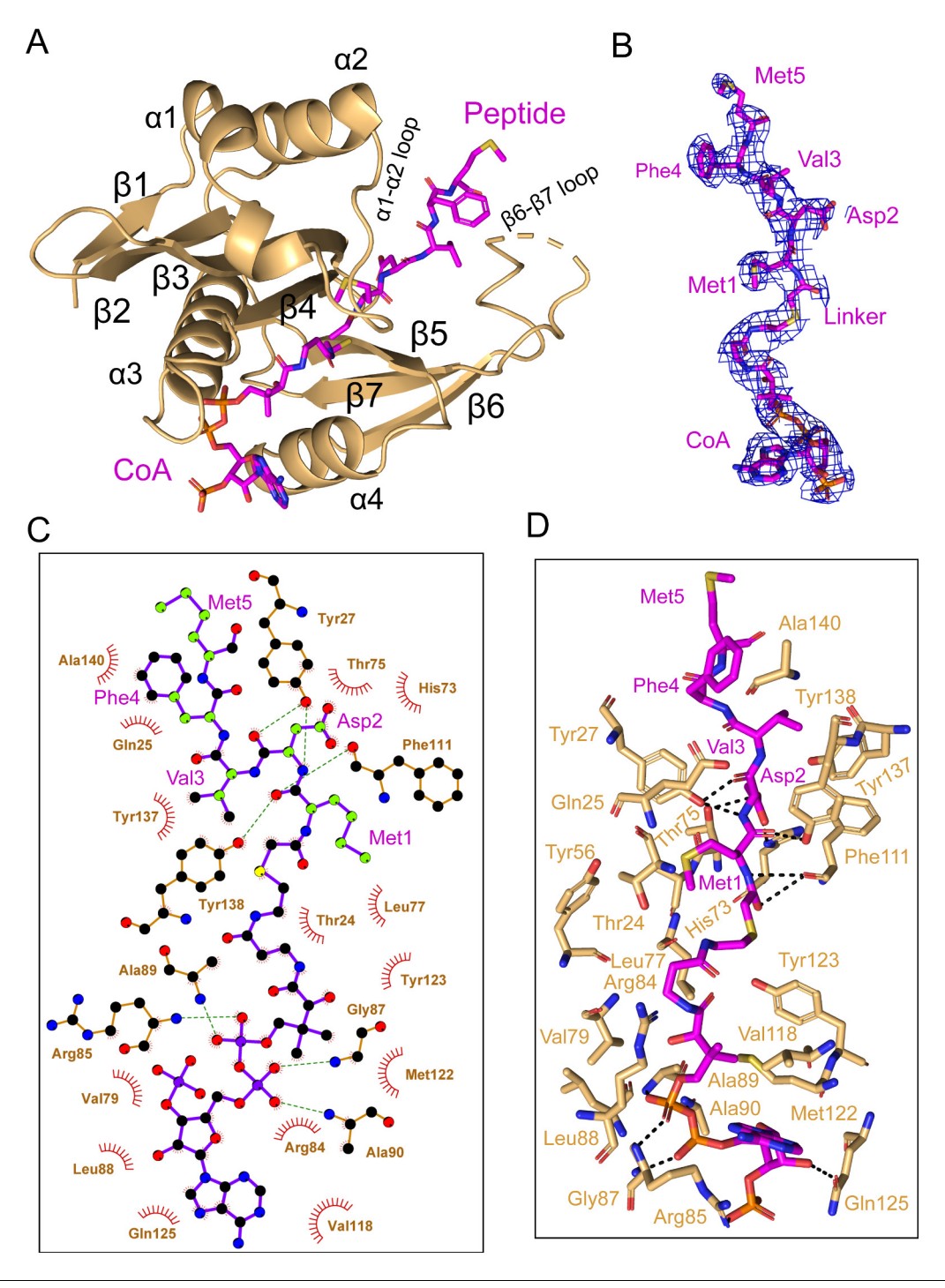

**Figure 4.** hNAA20 makes key CoA- and substrate peptide-interactions. (A) The structure of hNAA20 bound to the CoA-αSyn conjugate bound is shown in cartoon with corresponding secondary structures labeled. (B) The fit of the CoA-αSyn conjugate ligand in the EM density map. The contour level is 4.0 sigma. (C) Interaction between CoA-α Syn conjugate and hNAA20 residues is generated with LIGPLOT (*Laskowski and Swindells, 2011*). Hydrogen bonds are indicated by dashed green lines, and van der Waals interactions are indicated with red semicircles. (D) Highlighted polar and hydrophobic interactions between CoA-αSyn conjugate and the hNAA20 are depicted in the 3D view.

The online version of this article includes the following figure supplement(s) for figure 4:

**Figure supplement 1.** Sequence alignment of NAA20 homologs.

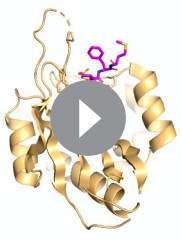

**Video 2.** Overall view of α-synuclein N-terminal interactions by NAA20. Amino acid sidechains that mediate hydrogen bond and van der Waals interactions with α-synuclein are highlighted in a cartoon model of NAA20.
https://elifesciences.org/articles/57491#video2

# Mutational analysis identifies key residues for hNatB catalysis and cognate substrate binding

To determine the functional importance of hNAA20 residues that appear to make important peptide or CoA substrate contacts in our model, we used an in vitro acetyltransferase assay to kinetically characterize WT and mutant hNatB proteins. Each mutant was purified to homogeneity and displayed identical gel filtration chromatography elution profiles (data not shown), indicating that they were all properly folded. We prepared alanine mutants of several residues involved in the CoA binding including Arg84, Arg85, Gly87, and Tyr123. Among them, R84A, R85A, and G87A did not show significant defects in overall protein catalytic function (*Table 1*). However, a Y123A mutant nearly abolished protein activity, with a 95% loss of protein activity, affecting both $k_{cat}$ and $K_m$ (*Table 1*). To further interrogate the properties of this residue, we prepared a Y123F mutant which features a similar aromatic bulky sidechain but not the polar p-hydroxyl group. We observed that Y123F displayed a similar ~88% loss of $k_{cat}$, but had a negligible effect on the peptide $K_m$ (*Table 1*). These data suggested that the Tyr123 hydroxyl group is critical for catalysis but not required for substrate binding, while the aromatic ring of Tyr123 plays a role in peptide substrate binding. Given that the hydroxyl group of Tyr123 is about 3.5 Å from the sulfur atom of the CoA-αSyn conjugate and 6.3 Å away from the αSyn N-terminus, it is in a position to play a role as a general base or acid for catalysis, potentially through an intervening water molecule (*Figure 5A*). This is analogous to the proposed general base role of Tyr73 as a general base for hNAA50 catalysis (*Liszczak et al., 2011*; see Discussion).

We also prepared alanine substitutions for residues that appeared to play important roles in αSyn binding: Glu25, Tyr27, His73, Tyr137, and Tyr138. We were surprised to find that mutations of hNatB residues that mediated backbone hydrogen bond interactions, Y27A and Y138A, had relatively modest effects on αSyn peptide NTA with Y27A showing ~twofold higher $K_m$ and Y138A showing ~twofold reduced $K_{cat}$, together suggesting that sidechain contacts might dominate the binding energy (*Table 1*). Consistent with this, and our structural observations, we found that H73A produced a ~90% reduction in activity (*Table 1*). This correlates with the importance of the His73 hydrogen bond and van der Waals contacts with αSyn-Asp2. Of note, other cognate hNatB sidechain residues at position 2, Glu, Gln, and Asn would also be well positioned to form hydrogen

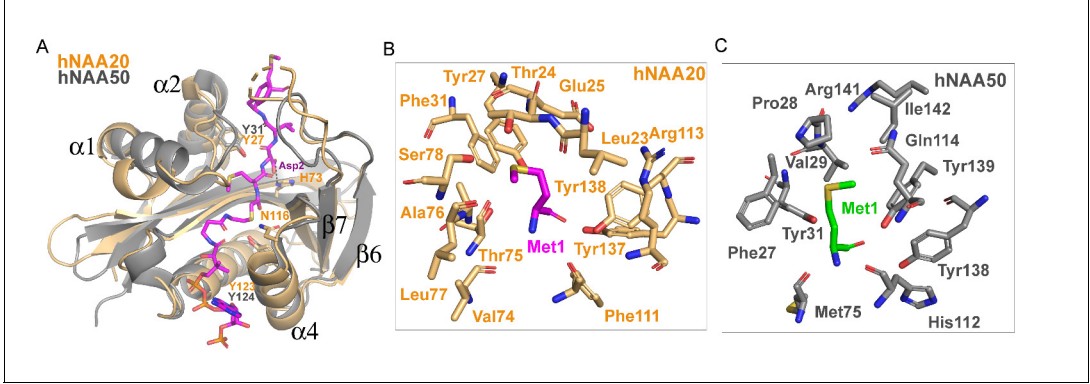

**Figure 5.** Structural comparison between hNAA20 and hNAA50. (**A**) Bi-substrate inhibitor-bound hNAA20 (light orange) is shown superimposed with hNAA50 (gray, PDB: 3TFY). H73, N116, and Y123 (sticks), mediate important functional roles in hNatB catalysis. (**B**) Residues forming the Met1 binding pocket of hNAA20 are depicted. (**C**) Residues forming the Met1 binding pocket of hNAA50 are depicted.

bonds with His73. Together, hNAA20-His73 appears to play a critical role in cognate substrate recognition by hNatB. hNatB-Asn116 is a highly conserved NatB residue (*Figure 4—figure supplement 1*) that caps the hNAA20 α4 helix, which also harbors the putative catalytic residue, Tyr123, and is also in position to make a water-mediated hydrogen bond with to CoA pantetheine nitrogen (*Figure 5A*). This observation suggests that Asn116 could play an important functional role. To test this, we prepared and evaluated an N116A mutant and, consistent with our hypothesis, Ans116, we found that this mutant leads to ~90% loss in activity (*Table 1*).

Taken together, our structural and mutational analysis of hNatB highlight the functional importance of hNatB-NAA20 residues His73, Asn116, and Tyr123 in hNatB-mediated N-terminal acetylation. While Tyr123 appears to play a critical catalytic role, potentially as a general base and/or acid for catalysis; His73 appears to play an important role in the recognition of substrate residue two and Asn116 likely plays a structural role (*Figure 5A*). Each of these residues could employ a bridging water molecule to mediate their functional roles, although these putative water molecules are not visible at the current resolution of our structure.

## Discussion

Since the identification of hNatB more than a decade ago, many studies have shown that it N-terminally acetylates important proteins such as actin, tropomyosin, CDK2, and α-Syn, and its function has connections to diseases such as hepatocellular carcinoma and Parkinson's disease (*Starheim et al., 2008*; *Polevoda et al., 2003*; *Ametzazurra et al., 2008*; *Neri et al., 2017*; *Halliday et al., 2011*; *Spillantini et al., 1998*). Despite its clear biological importance, hNatB-mediated NTA by hNatB remained poorly understood. Here we developed a CoA-αSyn conjugate hNatB inhibitor, determined the cryo-EM structure of CoA-αSyn inhibitor-bound hNatB, and carried out associated structure-guided mutagenesis and activity assays. This has led to the identification of functionally important differences with human NatA and *C. albicans* NatB. These studies have also provided evidence for important hNatB-specific elements responsible for αSyn recognition and N-terminal acetylation, providing direct implications for NatB recognition of other canonical substrate proteins.

Consistent with previous studies, we have demonstrated that hNatB acetylates a cognate 'MD' N-terminus and is unable to N-terminally acetylate non-cognate N-termini that are substrates for other NATs such as NatA, NatC, and NatE. We demonstrated for the first time that hNatB can acetylate an α-Syn peptide in vitro, directly linking hNatB to NTA of α-Syn. We have also demonstrated that αSyn peptides lacking the Met1 or both the Met1 and Asp2, do not serve as hNatB substrates, confirming the strict substrate specificity of hNatB. This is consistent with our structural model showing significant interactions between hNatB-NAA20 and both the first and second N-terminal residues of an αSyn peptide, with important but less extensive interactions with the third and fourth residues. This hierarchy of interactions likely explains how NatB enzymes can accommodate cognate substrates that diverge at positions three and four.

Here we have presented hNatB-αSyn interactions that can be used to rationalize the substrate specificity of hNatB: N-terminal sequences containing 'MD-', 'ME-', 'MN-', and 'MQ-'. αSyn-Met1 sits in a hydrophobic pocket that comfortably accommodates a methionine residue, whereas shorter sidechains or longer polar or charged sidechains would fit poorly (*Figure 5B–C*). The nature of this binding pocket is similar to the described hNAA50 recognition of Met1 (*Liszczak et al., 2011*; *Figure 5B–C*). Although both of hNAA50 and hNAA20 can N-terminally acetylate peptides with Met at the first position, no overlapping activity has been observed. This can be rationalized based on the chemical properties of the second residue in the cognate peptide. We find that the αSyn-Asp2 sidechain forms hydrogen bonds with the hNAA20 sidechains His73 and Thr75. These polar residues in the peptide binding site of hNAA20 would likely serve as poor acceptors of the largely hydrophobic residues targeted by hNAA50. The hNatB substrate client profile featuring D-, E-, N-, or Q-residues in position two is consistent with the mechanisms of substrate recognition observed in the binding pocket for αSyn-Asp2. The aliphatic regions of each of these sidechains (D, E, N, Q) would all benefit from the extensive hNatB van der Waals interactions surrounding the aliphatic region for αSyn-Asp2 (Tyr27, His73, Thr75, Phe111, and Tyr138). This second residue would also form a hydrogen bond interaction with His73 and Thr75, which may accommodate the carboxyl sidechains of both the shorter D- and N- and longer E- and Q- sidechains. Notably, His73 and Thr75 are strictly conserved from yeast to man (*Figure 5* and *Figure 2—figure supplement 3*). In contrast, shorter

polar or nonpolar sidechains would less efficiently fill the pocket for residue two, while larger polar or charged sidechains would likely result in steric clashes. In agreement with this, we have demonstrated that H73A mutation has a severe impact on hNatB catalysis (*Table 1*).

The hNatB/CoA-αSyn structure has implications for the mode of hNatB catalysis. While previous studies have suggested that hNAA50-Tyr31 plays an important role in catalysis, mutation of the corresponding hNaa20 residue, Tyr27, had minimal effects on hNatB kinetic parameters. Strikingly, our mutational analysis has identified the functional importance of hNatB-NAA20 residues His73, Asn116, and Tyr123, although Tyr123 is the only residue that is in position to play a catalytic role (*Figure 5A*). Specifically, Tyr123 is in position to play a catalytic role, potentially as a general base and/or acid, through a bridging water molecule (although a water molecule is not visible at the current resolution). Interestingly, hNAA50 contains a tyrosine residue at the same position (hNAA50-Tyr124), although the mechanistic significance of this tyrosine residue has not yet been described (*Liszczak et al., 2011*). It would be of interest to determine if hNAA50-Tyr124 also plays an important catalytic role, similar to the corresponding hNAA20-Tyr123 of hNatB.

The biological importance of hNatB and its connection to various disease processes highlights it as an important target for probe and inhibitor development. Indeed, a recent study highlights hNatB as a therapeutic target for αSyn toxicity (*Vinueza-Gavilanes et al., 2020*). Our development of a CoA-αSyn conjugate bisubstrate with an $IC_{50}$ of ~1.6 μM represents a step in this direction, although the structural information provided here could further aid to the rational development of more drug-like hNatB inhibitors with possible therapeutic applications.

# Materials and methods

## Key resources table

| Reagent type (species) or resource | Designation | Source or reference | Identifiers | Additional information |
|---|---|---|---|---|
| Recombinant DNA reagent | pFASTBac HTA-hNAA20(1-163) (plasmid) | This paper | | Protein expression plasmid for *hNAA20 (in Sf9 cells)* and found in the Materials and methods section of this paper |
| Recombinant DNA reagent | pFASTBac HTA-6HIS-TEV-hNAA25(1-972) (plasmid) | This paper | | Protein expression plasmid for *hNAA25 (in Sf9 cells)* and found in the Materials and methods section of this paper |
| Cell line (*Spodoptera frugiperda*) | Sf9 cells | ThermoFisher | cat #12659017 | For protein expression |
| Peptide, recombinant protein | MVDF peptide | GenScript | $NH_2$-MDVFMKGRW GRPVGRRRRP-COOH | |
| Peptide, recombinant protein | 'SASE' peptide | GenScript | $NH_2$-SASEAGVRWG RPVGRRRRP-COOH | |
| Peptide, recombinant protein | 'MLGP' peptide | GenScript | $NH_2$-MLGPEGGRW GRPVGRRRRP-COOH | |
| Peptide, recombinant protein | 'SGRG'/H4 peptide | GenScript | $NH_2$-SGRGKGGKGL GKGGAKRHR-COOH | |
| Peptide, recombinant protein | 'MLRF' peptide | GenScript | $NH_2$-ML RFVTKRW GRPVGRRRRP-COOH | |
| Peptide, recombinant protein | 'DVFM' peptide | GenScript | $NH_2$-DVFMKGLRW GRPVGRRRRP-COOH | |

*Continued on next page*

*Continued*

| Reagent type (species) or resource | Designation | Source or reference | Identifiers | Additional information |
|---|---|---|---|---|
| Peptide, recombinant protein | 'VFMK' peptide | GenScript | $NH_2$-VFMKGLSRW GRPVGRRRRP-COOH | |
| Other | [$^{14}$C] Acetyl-CoA (4 mCi/mmol) | PerkinElmer Life Sciences | Cat#NEC313050UC | |
| Other | P81 Phosphocellulose squares | EMD Millipore | Cat#20–134 | |
| Software, algorithm | RELION | *Zivanov et al., 2018* | | |
| Software, algorithm | MotionCor2 | *Zheng et al., 2017* | | |
| Software, algorithm | COOT | *Emsley and Cowtan, 2004* | | |
| Software, algorithm | PHENIX | *Adams et al., 2010* | | |
| Software, algorithm | ResMap | *Kucukelbir et al., 2014* | | |
| Software, algorithm | Gctf | *Zhang, 2016* | | |
| Software, algorithm | Prism 5.0 | GraphPad | | |
| Software, algorithm | PyMOL | Schrodinger LLC | | |

## Protein expression and purification

hNAA20 with a C-terminal truncation (1–163 out of 178 residues) and full-length hNAA25 were cloned into two separate insect cell expression vectors pFASTBac HTA. hNAA20 was untagged, while hNAA25 contained a Tobacco-etch virus (TEV)-cleavable N-terminal 6xHis-tag. Human NatB complex (hNAA20$^{1-163}$/hNAA25$^{FL}$) was obtained by co-expressing these two plasmids in Sf9 (*S. frugiperda*) cells (ThermoFisher, cat# 12659017), and purified as described previously (*Gottlieb and Marmorstein, 2018*). Sf9 cells were grown to a density of $1 \times 10^6$ cells/mL and infected using the amplified hNAA20$^{1-163}$/hNAA25$^{FL}$ baculovirus to an MOI (multiplicity of infection) of 1–2. The cells were grown at 27°C and harvested for 48 hr post-infection by centrifugation. Cell pellets were resuspended in lysis buffer (25 mM Tris, pH 8.0, 300 mM NaCl, 10 mM Imidazole, 10 mM β-ME, 0.1 mg/mL PMSF, DNase, and complete, EDTA-free protease inhibitor tablet) and lysed by sonication. After centrifugation, the supernatant was isolated and passed over Ni-NTA resin (Thermo Scientific), which was subsequently washed with 10 column volumes of lysis buffer. Protein was eluted with a buffer with 25 mM Tris, pH 8.0, 300 mM imidazole, 200 mM NaCl, 10 mM β-ME, which was dialyzed into a buffer with 25 mM HEPES pH 7.5 50 mM NaCl 10 mM β-ME. Ion-exchange was carried out with an SP ion-exchange column (GE Healthcare) in dialysis buffer with a salt gradient (50–750 mM NaCl). Peak fractions were concentrated to ~0.5 mL with a 50 kDa concentrator (Amicon Ultra, Millipore), and loaded onto an S200 gel-filtration column (GE Healthcare) in a buffer with 25 mM HEPES, pH 7.5, 200 mM NaCl, and 1 mM TCEP. Proteins were aliquoted, snap-frozen in liquid nitrogen, and stored at −80°C for further use. Protein harboring mutations were generated with the QuickChange protocol (Stratagene) and obtained following the same expression and purification protocol as described for the wild-type protein. Primers synthesized for the generation mutant constructs are listed in *Supplementary file 1*.

## Acetyltransferase activity assays

All acetyltransferase assays were carried out at room temperature in a reaction buffer containing 75 mM HEPES, pH 7.5, 120 mM NaCl, 1 mM DTT as described (*Deng et al., 2020*; *Deng et al., 2019*). The 'MDVF' peptide substrate was based on the first seven amino acid of α-Synuclein ('MDVF'

peptide: NH$_2$-MDVFMKGRWGRPVGRRRRP-COOH; 'SASE' peptide: NH$_2$-SASEAGVRWGRPVG RRRRP-COOH; 'MLGP' peptide: NH$_2$-MLGPEGGRWGRPVGRRRRP-COOH; 'SGRG'/H4 peptide: NH$_2$-SGRGKGGKG LGKGGAKRHR-COOH; 'MLRF' peptide: NH$_2$-ML RFVTKRWGRPVGRRRRP-COOH; 'DVFM' peptide: NH$_2$-DVFMKGLRWGRPVGRRRRP-COOH; 'VFMK' peptide: NH$_2$-VFMKG LSRWGRPVGRRRRP-COOH; GenScript). Reactions were performed in triplicate. To determine steady-state catalytic parameters of hNatB with respect to acetyl-CoA, 100 nM hNatB was mixed with a saturating concentration of 'MDVF' peptide substrate (500 μM) and varying concentrations (1.95 μM to 1 mM) of acetyl-CoA ($^{14}$C-labeled, 4 mCi mmol$^{-1}$; PerkinElmer Life Sciences) for 10 min reactions. To determine steady-state catalytic parameters of hNatB with respect to peptide substrate, 100 nM hNatB was mixed with saturating concentrations of acetyl- CoA (300 μM, $^{14}$C-labeled) and varying concentrations of 'MVDF' peptide (1.95 μM to 1 mM) for 10 min. Reactions were quenched by adding the solution to P81 paper discs (Whatman). Unreacted acetyl-CoA was removed by washing the paper discs in buffer with 10 mM HEPES, pH 7.5, at least three times, each 5 min. The paper discs were then dried with acetone and transferred to 4 mL scintillation fluid for signal measurement (Packard Tri-Carb 1500 liquid scintillation analyzer). Data was fitted to a Michaelis–Menten equation in GraphPad Prism to calculate kinetic parameters. Kinetic parameters on mutants with respect to peptide were carried out in the same condition as for wild type, with 300 μM $^{14}$C labeled acetyl-CoA and varied peptide concentration (1.95 μM to 1 mM). All radioactive count values were converted to molar units with a standard curve created with known concentrations of radioactive acetyl-CoA added to scintillation fluid. GraphPad Prism (version 5.01), was used for all data fitting to the Michaelis–Menten equation. For IC$_{50}$ determination of the CoA-αSyn conjugate, 100 nM hNatB was mixed with 500 μM 'MVDF' peptide and 300 μM $^{14}$C labeled acetyl-CoA, and inhibitor concentrations were varied (0.23 μM to 13.44 μM). Data were fit to a sigmoidal dose-response curve with GraphPad Prism (version 5.01). Errors represent s.d. (n = 3).

## Cryo-EM data collection

For initial sample screening, 0.6 mg/mL fresh hNatB sample with three-molar excess bisubstrate was used. hNatB particles on these grids exhibited a severe preferred orientation, which generated an incorrect 3D initial model (data not shown). To solve this issue, 1 μL of 0.05% NP-40 was mixed with 20 μL of hNatB (4 mg/mL). 3 μL of this sample was applied to glow-discharged Quantinfoil R1.2/1.3 holey carbon support grids, blotted and plunged into liquid ethane, using an FEI Vitrobot Mark IV. An FEI TF20 was used for screening the grids and data collection was performed with a Titan Krios equipped with a K3 Summit direct detector (Gatan), at a magnification of 105,000×, with defocus values from −0.1 to −2.0 μm. Each stack was exposed in super-resolution mode with a total dose of 45 e$^-$/Å$^2$, resulting in 35 frames per stack. Image stacks were automatically collected with Latitude software (Gatan, Inc).

## Cryo-EM data processing

Original image stacks were summed and corrected for drift and beam-induced motion at the micrograph level using MotionCor2 (*Zheng et al., 2017*), and binned twofold, resulting in a pixel size of 0.83 Å/pixel. Defocus estimation and the resolution range of each micrograph were performed with Gctf (*Zhang, 2016*). About 3000 particles were manually picked to generate several rough 2D class averages. Representative 2D classes were used to automatically pick ~1,927,673 particles from 5281 micrographs in Relion 3.0 (*Kimanius et al., 2016*; *Zivanov et al., 2018*). All particles were extracted and binned to accelerate the 2D and 3D classification. After bad particles were removed by 2D and 3D classification, 982, 420 particles were used for auto-refinement and per-particle CTF refinement. After refinement, a mask was created in Relion with an initial binarization threshold of 0.005, covering the protein complex and extending the binary map and soft-edge by 12 pixels. The map was sharpened with the created mask by estimating B-factor automatically in Relion. The final map was refined to an overall resolution of 3.46 Å, with local resolution estimated by Resmap (*Kucukelbir et al., 2014*). We attempted particle polishing on this data set but this surprisingly resulted in artifactual density in the resulting map. We believe that this was due to some small defects in the K3 camera during data collection, which corrupted the particle polishing process. We, therefore, did not perform particle polishing on this data set. Raw micrographs were deposited in EMPIAR with access ID of EMPIAR-10477.

## Cryo-EM model building and refinement

The hNatB atomic model was manually built de novo using the program COOT (*Emsley and Cowtan, 2004*) according to the cryo-EM map, with the guidance of predicted secondary structure and bulky residues such as Phe, Tyr, Trp, and Arg. The first two alpha helices of hNAA25 were built as poly-alanine due to the lack of tracible density in the 3D map. The complete model was then refined by real-space refinement in PHENIX (*Adams et al., 2010*). All representations of cryo-EM density and structural models were performed with Chimera (*Pettersen et al., 2004*) and PyMol (https://pymol.org/2/). The sequence alignments with secondary structure display were created by ESPript 3.0 (*Robert and Gouet, 2014*). hNAA25 TPR predictions were performed using the TPRpred server (*Karpenahalli et al., 2007*; *Zimmermann et al., 2018*; https://toolkit.tuebingen.mpg.de/#/tools/tprpred). The surface area calculation was performed using PDBePISA (*Krissinel and Henrick, 2007*; Proteins, Interfaces, Structures, and Assemblies; http://www.ebi.ac.uk/pdbe/pisa/).

## Acknowledgements

This work was supported by NIH grant R35 GM118090 awarded to RM and R01 NS103873 awarded to EJP. BP thanks the University of Pennsylvania for support through a Dissertation Completion Fellowship. We acknowledge the support of the Perelman School of Medicine, University of Pennsylvania DNA Sequencing Core Facility and D Schultz and E Dean from the University of Pennsylvania High Throughput Screening Core Facility for providing the Sf9 cells expressing hNatB for this study. We thank Dr. Zuo Biao and Dr. Sudheer Molugu from the University of Pennsylvania Electron Microscopy Resource Lab for help with initial cryo-grid screening; and Dr. Darrah Johnson-McDaniel from the Beckman Center for Cryo-EM at the University of Pennsylvania for technical assistance on data collection. We also thank Dr. Xuepeng Wei for help in data analysis and discussion.

## Additional information

### Funding

| Funder | Grant reference number | Author |
| --- | --- | --- |
| National Institutes of Health | R35 GM118090 | Ronen Marmorstein |
| National Institutes of Health | R01 NS103873 | James Petersson |
| National Institutes of Health | P01 AG031862 | Ronen Marmorstein |

The funders had no role in study design, data collection and interpretation, or the decision to submit the work for publication.

### Author contributions

Sunbin Deng, Conceptualization, Data curation, Formal analysis, Investigation, Methodology, Writing - original draft; Buyan Pan, Conceptualization, Data curation, Formal analysis, Investigation, Methodology, Writing - review and editing; Leah Gottlieb, Conceptualization, Data curation, Investigation, Methodology, Writing - review and editing; E James Petersson, Conceptualization, Data curation, Supervision, Funding acquisition, Project administration, Writing - review and editing; Ronen Marmorstein, Conceptualization, Resources, Formal analysis, Supervision, Funding acquisition, Project administration, Writing - review and editing

### Author ORCIDs

Sunbin Deng (iD) https://orcid.org/0000-0001-7798-4317
Ronen Marmorstein (iD) https://orcid.org/0000-0003-4373-4752

### Decision letter and Author response

Decision letter https://doi.org/10.7554/eLife.57491.sa1
Author response https://doi.org/10.7554/eLife.57491.sa2

## Additional files

### Supplementary files
• Supplementary file 1. Sequence of primers for preparing mutants. Both the forward and reverse primers for each mutant is indicated.

• Transparent reporting form

### Data availability
Cryo-EM data submissions to the Protein Data Bank (PDB code 6VP9), Electron Microscopy Data Bank (EMD code 21307) and Electron Microscopy Public Image Archive (EMPIAR-10477).

The following datasets were generated:

| Author(s) | Year | Dataset title | Dataset URL | Database and Identifier |
|---|---|---|---|---|
| Deng S, Marmorstein R | 2020 | Cryo-EM structure of human NatB complex | http://www.rcsb.org/structure/6VP9 | RCSB Protein Data Bank, 6VP9 |
| Deng S, Marmorstein R | 2020 | cryo-EM data for NatB complex | http://www.ebi.ac.uk/pdbe/entry/emdb/EMD-21307 | Electron Microscopy Data Bank, EMD-21307 |
| Deng S, Marmorstein R | 2020 | cryo-EM data for NatB comple | https://www.ebi.ac.uk/pdbe/emdb/empiar/entry/10477/ | Electron Microscopy Public Image Archive, 10477 |

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
