## [Decision Letter]

**Acceptance summary:**

This work reports a high resolution cryoEM structure of human NatB, whose N-terminal acetylation activity has been linked to both Parkinson's disease and hepatocellular carcinoma. Compared with other NAT complexes, little is known about hNatB. These studies reveal functional features of hNatB, key determinants for α-synuclein N-terminal acetylation, and important residues for substrate-specific recognition and acetylation. These results have implications for the development of drug-like hNatB inhibitors with possible therapeutic applications.

**Decision letter after peer review:**

Thank you for submitting your article "Molecular Basis for N-terminal Α-Synuclein Acetylation by Human NatB" for consideration by *eLife*. Your article has been reviewed by two peer reviewers, and the evaluation has been overseen by a Reviewing Editor and Huda Zoghbi as the Senior Editor. The following individual involved in review of your submission has agreed to reveal their identity: Sjors HW Scheres (Reviewer #1).

The two reviewers were positive about your submission, and we will be willing to accept it for publication subject to satisfactory responses to their comments below. N-terminal acetylation of α-synuclein is an area of interest. It is, for example, known to increase the affinity of α-synuclein for lipid membranes; in this context, the recent paper by Runfola et al. should be discussed.

Reviewer #1:

This paper describes a cryo-EM structure of human NatB in complex with a CoA-peptide with the N-terminal sequence of human α-synuclein, as well as complementing binding/activity assays with mutants that were designed based on the structure. The cryo-EM structure was done to good standards and the paper is very well written. I am therefore enthusiastic about publication in *eLife* and have only a few relatively minor comments:

"We therefore incorporated this sequence into a peptide substrate named "MDVF" for an in vitro acetyltransferase assay (see Materials and methods for the full sequence)." I thought this was a 4-residue peptide, but later found out it was longer. Why not just put the entire sequence here in the main text for improved clarity?

"among where the predicted first and second α-helices were built as poly-alanine due to a lack of resolvable side-chain density (Figure 4—figure supplement 1)." I had hoped to see some of the bad density for these helices in a supplemental figure. Actually, inspection of the map reveals that density is weak up to residue 146. This points to disorder or structural flexibility in this region, which should be explicitly mentioned in the main text. Did the authors try to perform focussed classification with signal subtraction (possibly keeping the orientations fixed) to select a subset of the 1M particles that would be better ordered in this region? Might be worth a try, as it may also provide insights in the nature of the flexibility in this region.

It would be good to have a figure of the cryo-EM density of the entire complex in a panel in one of the main figures. Currently, there are only a few panels with local zoomed-in views f the reconstructed cryo-EM density.

FSC curves between the refined PDB model and the cryo-EM are missing. Also, to show that the map was not overfitted, FSC_work and FSC_free curves (where one refines a perturbed model against one half-map and then calculates FSCs of that model against both half-maps) should be included.

The ResMap doesn't look right: I strongly doubt the local resolution in the N-terminal part of hNAA25 is 4.5A: it looks a lot worse. Perhaps it's just a matter of choosing the colour scale, or perhaps the authors could try RELION's own implementation of local resolution estimation.

"We demonstrated for the first time that hNatB can acetylate an α-Syn

peptide in vitro, directly linking hNatB to Parkinson disease." That is too strong: the structure does not provide a direct link to PD.

Cryo-EM processing: why was Bayesian polishing not performed? Also, with ~1M particles left, it could be that more classification after the polishing could further improve the map.

Cryo-EM processing: How was map post-processing (map sharpening) performed?

Reference Kimanius et al., 2016: better use the RELION-3 reference by Zivanov et al., 2018.

Reviewer #2:

Deng et al. present the structure and functional analysis of the human NatB protein, bound to CoA with an αSyn fragment. I only comment on the methodological aspects of the manuscript.

The method description in the main text is rather short. A Titan Krios with K3 was used, and Table 2 lists parameters. However, these are not sufficient to verify the quality of the work. For example, which software was used for data collection (EPU or SerialEM or other)? How many frames were used per video? What mode of the K3 was used (CDC or Counting)? The magnification is specified as 105kx, but with the 5micrometer pixel size of the K3 that would correspond to a pixel size on the sample of 0.476A at the sample level. However, the Table 2 lists a pixel size of 0.83A per pixel. Is that physical pixel or super-resolution pixel? In any case, the magnification value is probably wrong, and may be a historically originated software bug in the used Titan Krios? Or is that magnification valid for the Flu-screen in the Titan, which has nothing to do with the K3?

How many KDa is the molecular weight of the determined structure?

In addition to the model in the PDB, the raw data should be deposited in the EMPIAR and the reconstructed map should be deposited in the EMDB.

[Editors' note: further revisions were suggested prior to acceptance, as described below.]

Thank you for resubmitting your work entitled "Molecular Basis for N-terminal Α-Synuclein Acetylation by Human NatB" for further consideration by *eLife*. Your revised article has been evaluated by Huda Zoghbi (Senior Editor) and a Reviewing Editor after consultation with the reviewers.

The manuscript has been improved but there are some remaining issues that need to be addressed before acceptance, as outlined below:

Please confirm that in Figure 2—figure supplement 1, panel D, the model used for the 2 half-map curves was actually refined in one of the two half-maps, and then used for FSC against both half-maps. They should also indicate which of the two half-maps it was refined in. Currently, it reads (which indeed seems to be the case) that these are not the requested FSC_work and FSC_test curves, but that instead they have just used the model from the refinement against the final map to calculate a FSC curve against the two half-maps. That is not informative. The key is refinement in one half-map (which may be overfitted, but this would then show up in a lower FSC curve with the other half-map). This needs to be resolved.

---

## [Author Response]

The two reviewers were positive about your submission, and we will be willing to accept it for publication subject to satisfactory responses to their comments below. N-terminal acetylation of α-synuclein is an area of interest. It is, for example, known to increase the affinity of α-synuclein for lipid membranes; in this context, the recent paper by Runfola et al. should be discussed.

In the Introduction, a sentence is added to summarize the results from the recent manuscript by Runfola et al., which demonstrates the importance of N-terminal acetylation on the lipid membrane binding properties of a-synuclein.

Reviewer #1:This paper describes a cryo-EM structure of human NatB in complex with a CoA-peptide with the N-terminal sequence of human α-synuclein, as well as complementing binding/activity assays with mutants that were designed based on the structure. The cryo-EM structure was done to good standards and the paper is very well written. I am therefore enthusiastic about publication in eLife and have only a few relatively minor comments:"We therefore incorporated this sequence into a peptide substrate named "MDVF" for an in vitro acetyltransferase assay (see Materials and methods for the full sequence)." I thought this was a 4-residue peptide, but later found out it was longer. Why not just put the entire sequence here in the main text for improved clarity?

Full sequences for the peptide substrates used in the study are now indicated in the main text.

"among where the predicted first and second α-helices were built as poly-alanine due to a lack of resolvable side-chain density (Figure 4—figure supplement 1)." I had hoped to see some of the bad density for these helices in a supplemental figure. Actually, inspection of the map reveals that density is weak up to residue 146. This points to disorder or structural flexibility in this region, which should be explicitly mentioned in the main text. Did the authors try to perform focussed classification with signal subtraction (possibly keeping the orientations fixed) to select a subset of the 1M particles that would be better ordered in this region? Might be worth a try, as it may also provide insights in the nature of the flexibility in this region.

We added a sentence in the main text stating “The N terminal region (residue 1-164) displays relatively weak EM density compared to other regions, suggesting that it is relatively flexible (Figure 2—figure supplement 2).”

While we agree that further focused classification might improve the local resolution of the N-terminal region, we have chosen not to do this since this region does not appear to mediate biologically important interactions in our structure and would therefore not alter the main conclusions of our study. Nonetheless, the raw cryo-EM data have been deposited in the EMPIAR database so other researchers can access and analyze this dataset further if an interest arises.

It would be good to have a figure of the cryo-EM density of the entire complex in a panel in one of the main figures. Currently, there are only a few panels with local zoomed-in views f the reconstructed cryo-EM density.

We now show the fit of the complex into the cryo-EM density in Figure 2—figure supplement 2, panel D. This figure also illustrates the relatively poor density of the N-terminal region of hNatB.

FSC curves between the refined PDB model and the cryo-EM are missing. Also, to show that the map was not overfitted, FSC_work and FSC_free curves (where one refines a perturbed model against one half-map and then calculates FSCs of that model against both half-maps) should be included.

We have now calculated the FSC curves with the refined PDB model against each half map and the summed map, and show these curves in Figure 2—figure supplement 1 panel D.

The ResMap doesn't look right: I strongly doubt the local resolution in the N-terminal part of hNAA25 is 4.5A: it looks a lot worse. Perhaps it's just a matter of choosing the colour scale, or perhaps the authors could try RELION's own implementation of local resolution estimation.

We thank the review for pointing this out. The previous local resolution map likely did not seem correct because most of the regions of NAA25, except for the N-terminal region (1-164), has a resolution beyond 4.5 Å. We have re-run a ResMap in Relion with different color scale and have illustrated a new local resolution map in Figure 2—figure supplement 1, panel B.

"We demonstrated for the first time that hNatB can acetylate an α-Synpeptide in vitro, directly linking hNatB to Parkinson disease." That is too strong: the structure does not provide a direct link to PD.

We have softened this statement to “directly linking hNatB to NTA of α-Syn”

Cryo-EM processing: why was Bayesian polishing not performed? Also, with ~1M particles left, it could be that more classification after the polishing could further improve the map.

We attempted particle polishing on this dataset, but this surprisingly resulted in artifactual density in the resulting map. We believe that this was due to some small defects in the K3 camera during data collection, which corrupted the particle polishing process. We therefore were unable to exploit particle polishing to improve the resolution of our structure. Instead, two rounds of CTF refinement were performed to achieve the current resolution for model building. This is described in the Materials and methods section of the manuscript.

Cryo-EM processing: How was map post-processing (map sharpening) performed?

This is now explained in the Materials and methods section as follows: After refinement, a mask was created in Relion with an initial binarization threshold of 0.005, covering the protein complex and extending the binary map and soft-edge by 12 pixels. The map was sharpened with the created mask by estimating B-factor automatically in Relion.

Reference Kimanius et al., 2016: better use the RELION-3 reference by Zivanov et al., 2018.

This reference (Zivanov et al., 2018) is now added.

Reviewer #2:Deng et al. present the structure and functional analysis of the human NatB protein, bound to CoA with an αSyn fragment. I only comment on the methodological aspects of the manuscript.The method description in the main text is rather short. A Titan Krios with K3 was used, and Table 2 lists parameters. However, these are not sufficient to verify the quality of the work. For example, which software was used for data collection (EPU or SerialEM or other)? How many frames were used per video? What mode of the K3 was used (CDC or Counting)? The magnification is specified as 105kx, but with the 5micrometer pixel size of the K3 that would correspond to a pixel size on the sample of 0.476A at the sample level. However, the Table 2 lists a pixel size of 0.83A per pixel. Is that physical pixel or super-resolution pixel? In any case, the magnification value is probably wrong, and may be a historically originated software bug in the used Titan Krios? Or is that magnification valid for the Flu-screen in the Titan, which has nothing to do with the K3?

We have now added more detail of how the data were collected in the Materials and methods section. With regard to the pixel size, we now indicate that while we used the super resolution mode with a pixel size of 0.415 Å /pixel, we binned the data twofold to 0.83 Å /pixel when motion correction was performed and all further data processing was based on the motion-corrected images with 0.83 Å /pixel.

How many KDa is the molecular weight of the determined structure?

We now indicate in the Introduction section that hNatB is ~130 KDa.

In addition to the model in the PDB, the raw data should be deposited in the EMPIAR and the reconstructed map should be deposited in the EMDB.

We have now deposited the raw data (Raw videos, picked-particle star file and others) into EMPIAR with access ID of EMPIAR-10477. This information is now added in Table 2. The model and map will be released when the manuscript is published, with access ID of PDB 6VP9, or EMD-21307.

[Editors' note: further revisions were suggested prior to acceptance, as described below.]

Please confirm that in Figure 2—figure supplement 1, panel D, the model used for the 2 half-map curves was actually refined in one of the two half-maps, and then used for FSC against both half-maps. They should also indicate which of the two half-maps it was refined in. Currently, it reads (which indeed seems to be the case) that these are not the requested FSC_work and FSC_test curves, but that instead they have just used the model from the refinement against the final map to calculate a FSC curve against the two half-maps. That is not informative. The key is refinement in one half-map (which may be overfitted, but this would then show up in a lower FSC curve with the other half-map). This needs to be resolved.

We thank the reviewer for pointing out this mistake. We have now refined the model in one of the independent half maps (in the new figure, we actually refined the model in half map 2), and then used the newly refined model to calculate the FSC curves against both independent half maps. The figure is now replaced with a new one, and the figure legend is updated to indicate which half map was used to refine in, to distinguish FSC_work and FSC_test.